# Calibration and Implementation of a Dynamic Energy Balance Model to Estimate the Temperature in a Plastic-Covered Colombian Greenhouse

Gloria Alexandra Ortiz [1], Adrian Nicolas Chamorro [1], John Fabio Acuña-Caita [2], Irineo L. López-Cruz [3] and Edwin Villagran [4],*

1    Corporación Colombiana de Investigación Agropecuaria—Agrosavia, Centro de Investigación Tibaitata, Km 14, Vía Mosquera-Bogotá, Mosquera 250040, Colombia; glaortizro@unal.edu.co (G.A.O.); achamorro@agrosavia.co (A.N.C.)
2    Departamento de Ingeniería Civil y Agrícola, Facultad de Ingeniería, Universidad Nacional de Colombia, Cra. 30 No. 45-03, Bogotá 111321, Colombia; jfacunac@unal.edu.co
3    Posgrado en Ingenieria Agricola y Uso Integral del Agua, Universidad Autónoma Chapingo, Km. 38.5 Carretera Mexico-Texcoco, Texcoco de Mora C.P. 36230, Mexico; ilopez@correo.chapingo.mx
4    Corporación Colombiana de Investigación Agropecuaria—Agrosavia, Sede Central, Km 14, Vía Mosquera-Bogotá, Mosquera 250040, Colombia
*    Correspondence: evillagran@agrosavia.co; Tel.: +57-1-4227-300 (ext. 1239)

**Abstract:** Modeling and simulation have become fundamental tools for the microclimatic analysis of greenhouses under various climatic conditions. These models allow precise control of the climate inside the structures and the optimization of their performance under any situation. In Colombia, the availability of energy balance models adapted to local greenhouses and their climate is limited, which affects the decision-making of both technical advisors and growers. This study focused on calibrating and evaluating a dynamic energy balance model to predict the thermal behavior of an innovative type of plastic-covered greenhouse designed for the Bogotá savanna. The selected model considers fundamental heat and mass transfer processes, incorporating parameters that depend on the architecture of the structure and local climatic conditions, making it suitable for protected agriculture in Colombia. The results of the post-calibration evaluation showed that the model is highly accurate, with a temperature prediction efficiency close to 86%. This ensures that the model can accurately predict the thermal behavior of the greenhouse being evaluated. It is important to note that the model can also anticipate phenomena characteristics of Colombian greenhouses, such as thermal inversion. This advance has become a valuable tool for decision-making in protected agriculture in the region.

**Keywords:** physical modeling; energy balance; temperature prediction; greenhouse microclimate

## 1. Introduction

The use of greenhouses and protected agriculture structures, with different techno-logical levels, has been worldwide since the 1980s, especially to produce ornamental, horticultural, aromatic, and medicinal plants [1,2]. It is important to mention that both the yield and the final quality of these crops are strongly influenced by the behavior of the environmental conditions generated inside the greenhouse. One of the most relevant of these conditions is the temperature since it influences the growth and development processes of the plants; moreover, it also represents the main source of energy consumption inside the greenhouse [3].

The temperature inside a greenhouse can be determined through microclimate monitoring, using sensors at various points within the structure. This practice allows the collection of key information related to the agronomic behavior of the cultivated species [4]. In addition, when this information on temperature behavior is collected with the use of

multiple sensors at different discrete points in the greenhouse, it can be analyzed using geostatistical techniques that allow the spatial variability in a specific plane inside the greenhouse to be determined. This makes it possible to identify areas that are subject to significant thermal gradients, which may be related to low yields and a higher incidence of pests and diseases [5,6]. Another widely used technique to determine the temperature in greenhouses is numerical simulation using computational fluid dynamics [7–9].

In addition, the use of modeling techniques has been another tool used to determine temperature and other microclimatic variables. These models have been proposed by numerous researchers who have worked on this subject over the last seven decades [10–12]. The typology of implemented models can be divided into physical models based on mass and energy balance equations and black box models using learning algorithms [13]. Physical models are built on mathematical relationships describing heat and mass exchange processes. They also describe the thermodynamic relationships that occur inside the greenhouse due to solar energy and the physical and physiological processes of the plants [14]. Despite the limitations and requirements of experimental equipment, these physical models have gained great importance in thermal evaluation and energy optimization for different types of greenhouses [15].

One of the first mathematical models used to determine the energy performance of greenhouses was documented by Businger [16]. Later, in the 2000s, the microclimatic modeling of protected agriculture structures became an area of knowledge with considerable research and didactic interest to understand the physical functioning of these agricultural production environments [17]. Simultaneously, this analysis technique was implemented as a fundamental tool for decision-making in the control and microclimatic management of greenhouses, as well as for the design of prototypes and climate conditioning equipment [10]. Currently, protected or greenhouse agriculture is one of the most widely used agricultural production tools to sustainably intensify food production and as a production mechanism to promote better levels of food security in various regions of the world [18]. However, this industry or production alternative also has important challenges related to the optimization and efficient use of natural and energy resources in greenhouses or other types of structures, requiring the sustainability of these production systems [19,20].

In accordance with the above, and because of technological and computational advances that allow the development of data analysis and processing in short periods of time, currently, there are more than 140 physical models capable of describing and predicting the microclimatic environment within various types of greenhouses established worldwide. It should be noted that approximately 50% of these models have been generated in the last decade [10,21]. This large variety of generated models can be attributed to the particularities of greenhouses in each region, which require adaptations and the development of mathematical models specific to each structure [22].

In the case of the physical modeling of the greenhouse microclimate, there are descriptive and mechanistic models; the latter are also known as white box or process-based models. Process-based models provide the modeler with a deeper understanding of the system under analysis and allow the evaluation of hypothetical scenarios, which provides more information on the variable under study [23]. Within this group of process-based models, the most used in greenhouses are the models that consider the air inside the greenhouse as a perfectly agitated mixing tank environment. This implies that the spatial variability that may occur in the behavior of the parameter under analysis must be ignored [24]. These simulation models are excellent for accurately predicting energy flows and, thus, temperature inside greenhouses. This prediction is made based on local climatic characteristics as well as soil characteristics. The prediction models also include other greenhouse characteristics, such as the type of roofing material, the surface and volume of the structure, and the location and type of ventilation [25].

Several studies have been conducted around the world in which models have been implemented to predict the microclimatic conditions of greenhouses. Some of them serve as a design methodology for different types of greenhouses and in different climatic

conditions, such as the proposed model by Vanthoor et al. [26]. This model has been successfully validated in several greenhouses in Italy and Spain, especially in tomato crops. Likewise, this type of balance model to predict temperature has been used to generate optimal control over the energy consumption of a greenhouse, seeking to optimize the cooling and heating practices together with the management of ventilation [27]. In terms of energy, modeling and experimental tests have shown that the exchange between the greenhouse floor and the outside environment has a strong impact on the temperature behavior. Likewise, it has been found that the layers located below 50 cm of soil are the best area to install thermal storage elements for greenhouse climate control [28]. At the regional level, a study was carried out by Salazar-Moreno et al. [29], in which a dynamic energy balance model was adjusted to predict the temperature in a greenhouse in Mexico, obtaining a prediction efficiency of more than 89%.

Accordingly, the objective of this work was to adjust and validate an energy balance model to determine the dynamic behavior of the temperature inside a new type of greenhouse with a plastic cover that was designed for the climatic conditions of the Colombian high tropics. The novelty of this work is that it includes equations that allow the management of the lateral and roof ventilation areas traditionally used in Colombia to be modeled. In addition, the model, being versatile, can be implemented in the different types of Colombian inveranderos just by making some changes in parameters related to the geometry of the structure to be analyzed.

## 2. Materials and Methods

### 2.1. Description of the Greenhouse

The analyzed greenhouse is located in the municipality of Mosquera Cundinamarca, in the Tibaitata research center of the Corporación Colombiana de Investigacion Agropecuaria—Agrosavia—with the following GPS coordinates: 4°41′43.89″; N 74°12′12.87″ W, and at an altitude of 2545 m above sea level. The greenhouse has a covered area of 20 m$^2$, projected on a side and front length of 4 and 5 m, respectively; the height above the gutter is 2.7 m and the height above the ridge of the roof ventilation is 5.8 m (Figure 1).

The greenhouse is equipped with an Agroclear polyethylene plastic cover, both in the roof and in the lateral and front areas. The natural ventilation system is composed of two lateral windows with a manually operated mechanical winch closure (3.6 m long and 2 m wide in dimension) and double roof ventilation (4 m long and 0.705 m wide). For the validation of the dynamic model, no climatic conditioning via heating or cooling was performed, and neither was any type of cultivation.

### 2.2. Dynamic Model of Energy Balance

The dynamic model used in this research without the presence of cultivation was structured using the energy balance model applied by various researchers in different regions of the world [30,31]. This model is described using the following ordinary differential equation:

$$\frac{dT_a}{dt} = \frac{1}{Vc_p\rho}(Q_s - Q_c - Q_v - Q_{sl} + Q_{co}) \tag{1}$$

where $T_a$ is the air temperature inside the greenhouse (°C), $dt$ is the unit of time change (s), $V$ is the volume of the greenhouse (m$^3$), $c_p$ is the specific heat of air (J kg$^{-1}$°C$^{-1}$), $\rho$ is the density of air (kgm$^{-3}$), $Q_s$ is heat gain due to solar radiation (Wm$^{-2}$), $Q_c$ is the heat exchange through the cover (Wm$^{-2}$), $Q_v$ (Wm$^{-2}$) is the loss of heat due to ventilation, $Q_{sl}$ is the heat loss through the soil (Wm$^{-2}$), and $Q_{co}$ is the heat gain due to condensation (Wm$^{-2}$). The heat gain due to solar radiation can be expressed as:

$$Q_S = Ir_{total} * \tau \tag{2}$$

where $Ir_{total}$ is the total radiation (Wm$^{-2}$) and $\tau$ is a coefficient of radiation transmission to the interior of the greenhouse, which represents the capacity of the roofing material to

allow the passage of radiation into the greenhouse and depends on factors such as the type of material used and its state of use. Heat loss through the greenhouse cover can be modeled with the following expression:

$$Q_c = \alpha_c \left( \frac{A_c}{A_s} \right) (T_i - T_o) \tag{3}$$

where $\alpha_c$ is the heat transfer coefficient of the roofing material ($\mathrm{Wm^{-2}{}^\circ C^{-1}}$) and represents the ability of the roofing material to transfer heat between the indoor ambient air and the outdoor air. $A_c$ and $A_s$ are the area of the roof including the walls and the area of the floor covered, respectively ($\mathrm{m^2}$); $T_i$ and $T_o$ are the indoor and outdoor air temperatures, respectively ($^\circ$C), whose difference drives the heat flow through the roof. As for the heat flow lost through ventilation, it is modeled using the following equation:

$$Q_v = \varphi_v * \rho * c_p * (T_i - T_o) \tag{4}$$

where $\varphi_v$ is the ventilation rate per unit of floor area ($\mathrm{m^3 m^{-2} s^{-1}}$); this ventilation rate can be calculated according to the relationship proposed by Baeza et al. [32]:

$$\varphi_v = \left( \frac{A_v}{2} \right) * C_d * \sqrt{C_w} * w_s + \varphi s_v \tag{5}$$

where $A_v$ is the ventilation area in ($\mathrm{m^2}$), $\varphi s_v$ is the ventilation rate generated by the infiltration per unit of floor area ($\mathrm{m^3 s^{-1}}$), $w_s$ is the external wind speed ($\mathrm{ms^{-1}}$), $C_w$ is the coefficient of ventilation due to wind, and $C_d$ is the discharge coefficient that is characteristic for each greenhouse and can be calculated for this type of greenhouse by using the following two equations:

$$C_d = F_0{}^{-0.5} \tag{6}$$

$$F_0 = 1.75 + 0.7 e^{-\lceil \frac{l_0}{w_0} \rceil / 32.5} \tag{7}$$

where $F_0$ is the pressure loss coefficient, and $l_0$ and $w_0$ are the length and width of the windows, respectively (m). For this specific case, $A_v$ is composed of the total lateral ventilation surface $2*A_{vS}$ and the roof ventilation surface $A_{vR}$; therefore, $A_v$ is calculated using the following equations:

$$A_v = 2 * A_{vS} + A_{vR} \tag{8}$$

$$A_{vS} = l_{vS} * w_{vS} \tag{9}$$

$$A_{vR} = l_{vR} * w_{vR} \tag{10}$$

where $l_{vS}$ and $l_{vR}$ are the lengths of the side and roof ventilation surfaces, respectively (m); $w_{vS}$ and $w_{vR}$ are the width of the side and roof ventilation surfaces, respectively (m). This modification is made because, during the period from 7:00 to 17:00 h, the greenhouse remains with the lateral ventilation areas open; moreover, during the night, in the period between 17:00 and 7:00 h, these areas remain closed. On the other hand, the infiltration ventilation rate is calculated with the following equation:

$$\varphi s_v = A_s * C_f \tag{11}$$

where $C_f$ is the infiltration coefficient ($m^3m^{-2}s^{-1}$). On the other hand, the heat loss through the soil can be obtained by means of the relation established by Van Ooteghem [33]:

$$Q_{sl} = \frac{k_s A_s (T_i - T_s)}{P_s} \qquad (12)$$

where $k_s$ is the coefficient of heat exchange through the soil ($Wm^{-1}{}^{\circ}C^{-1}$), and $P_s$ is the depth at which the soil temperature is measured and recorded (this depth is given in m). Finally, the heat gain due to condensation is modeled as established by Speetjens et al. [34] through the following mathematical relationships:

$$Q_{co} = \mu(\varphi_{co}) \qquad (13)$$

$$\varphi_{co} = k_{po} * \rho * A_c * (\beta - \beta_{sat}) \qquad (14)$$

$$\beta_{sat(T_C)} = 0.611 \left[ 1 + 1.414 \sin\left(5.82e^{-3}T_C\right) \right]^{8.827} \qquad (15)$$

$$\beta = \frac{0.611 HR_i * \beta_{sat(T_i)}}{100P} \qquad (16)$$

where $\varphi_{co}$ is the condensation conductance ($kgs^{-1}$), $k_{po}$ is the mass transfer coefficient that determines the condensation rate ($ms^{-1}$), and $\beta$ and $\beta_{sat}$ are the concentration of water vapor in the greenhouse and the concentration of saturated water vapor ($kgm^{-3}$). $T_C$ is the temperature of the cover ($^{\circ}C$), $HR_i$ is the relative humidity inside the greenhouse, which must be converted to absolute humidity ($g_{agua}m^{-3aire}$), and $P$ is the atmospheric pressure (hPa). The model was programmed, and the numerical simulations were developed in a MATLAB-SIMULINK (V. 9.13.0.219) programming environment using an academic license of this commercial software. The main parameters of the model were defined in this software, including input variables, simulation options, and output variables as explained in the work developed by Salazar-Moreno et al. [29].

### 2.3. Parameters of the Experimental Greenhouse

The parameters of the experimental greenhouse used in the energy balance model are detailed in Table 1.

**Table 1.** Parameters of the mathematical model of the greenhouse evaluated.

| Symbol | Description | Value | Units | Source |
|--------|-------------|-------|-------|--------|
| $V$ | Greenhouse volume | 86 | $m^3$ | Calculated |
| $A_s$ | Area of soil covered by the greenhouse | 20 | $m^2$ | Calculated |
| $A_c$ | Roof and wall area | 97.76 | $m^2$ | Calculated |
| $A_v$ | Ventilation surface (side and roof) | 20.04 | $m^2$ | Calculated |
| $\tau$ | Radiation transmission of the roof | 85 | % | Technical data sheet of the plastic |

**Table 1.** *Cont.*

| Symbol | Description | Value | Units | Source |
|--------|-------------|-------|-------|--------|
| $c_p$ | Specific heat of air at constant pressure | 1.006 | $\text{J kg}^{-1}\,^\circ\text{C}^{-1}$ | Calculated |
| P | Atmospheric pressure | 742.6 | hPa | Calculated |
| $\rho$ | Air density | 1.1 | $\text{kgm}^{-3}$ | Calculated |
| $\alpha_c$ | Heat transfer coefficient of the cover | 5 | $\text{W m}^{-2\circ}\text{C}^{-1}$ | [35] |
| $l_0$ | Length of ventilation surface | 3.6 | m | Calculated |
| $w_0$ | Ventilation surface width | 2 | m | Calculated |
| $C_d$ | Discharge coefficient | 0.64 | dimensionless | Calculated |
| $C_f$ | Infiltration coefficient | 0.008361 | $\text{m}^3\,\text{m}^{-2}\,\text{s}^{-1}$ | [36] |
| $C_w$ | Ventilation coefficient wind effect | 0.15 | dimensionless | [35] |
| $P_s$ | Depth at which soil temperature is estimated | 0.10 | m | Experimental |

## 2.4. Climate and Microclimate Data Collection

An automatic weather station with remote transmission WSC11-Thies was used to measure and record outdoor climate data. The technical description of each of the measurement sensors is summarized in Table 2.

Inside the greenhouse, an Aranet sensor and a data logging station were used to record air temperature. In addition, two sensors were used to measure and record soil and greenhouse cover temperature data, which were connected to an application generated using the Arduino open-source electronic prototyping platform (V. 2.2.1). The technical specifications of these sensors are detailed in Table 3. The air and soil temperature sensors were located in the central zone of the greenhouse at 1.0 m above ground level and 0.1 m deep, respectively, while the greenhouse cover sensor was located in the central zone just above the middle cross-section of the roof.

**Table 2.** Technical description of outdoor climate measurement sensors.

| Variable | Technical Sensor Data |
|----------|----------------------|
| Wind speed. | Type: Thermal anemometer, measuring range: 0 to 40 $\text{ms}^{-1}$, resolution: 0.1 $\text{ms}^{-1}$, accuracy: 0.1 $\text{ms}^{-1}$. |
| Wind direction. | Type: Thermal anemometer, measuring range: 1 to 360°, resolution: 1°, accuracy: ±10°. |
| Solar radiation. | Type: Silicium sensor, measuring range: 1 to 1300 $\text{Wm}^{-2}$, resolution: 1 $\text{Wm}^{-2}$, accuracy: ±10%. |
| Temperature. | Type: PT1000, measuring range: -30 to +60 °C, resolution: 0.1 °C, accuracy: ±1 °C. |
| Relative humidity. | Type: CMOS capacitive, measuring range: 0 to 100%, resolution: 0.1%, accuracy: ±10%. |

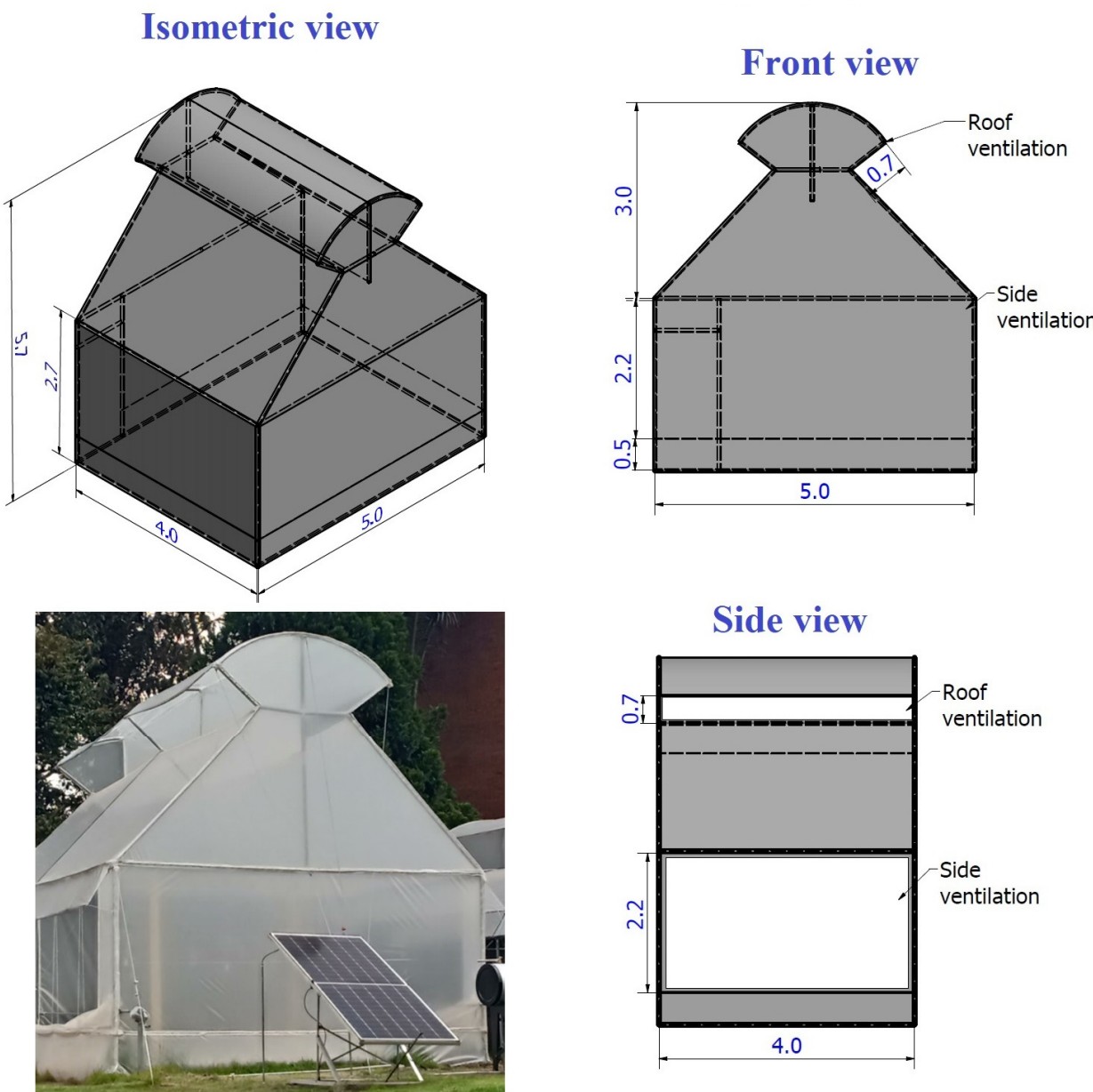

**Figure 1.** Architectural description of the evaluated greenhouse.

**Table 3.** Technical description of the microclimatic measurement sensors.

| Variable | Technical Sensor Data |
| --- | --- |
| Data logging station. | Aranet PRO licence versions select, Aranet PRO 50. |
| Air temperature. | Type: IP67 wireless and battery powered, Aranet IP 67, measuring range: −40 to +60 °C and 0 to 100%, resolution: 0.1 °C and 0.1%, accuracy: ± 0.3 °C and ±2%. |
| Greenhouse cover temperature. | Type: IR temperature sensor, MLX90614ESF, measuring range: −40 to 125 °C, resolution: 0.02 °C, accuracy: ±0.5 °C. |
| Soil temperature. | Type: K-type thermocouple MAX6675, measuring range: 0 to 800 °C, resolution: 0.25 °C, accuracy: ±1 °C. |

The data outside and inside the experimental greenhouse were collected during the period from 27 March to 1 April 2023. The frequency of data recording was every one minute, which resulted in a total of 8637 data for each variable. This data set was used for the calibration, evaluation, and numerical testing of the energy balance model proposed.

*2.5. Model Calibration and Implementation*

For the initial calibration of the model, we followed the methodology proposed in the work developed by Salazar-Moreno et al. [29]. This methodology consists of varying some of the specific parameters of the greenhouse's architecture, with the objective of obtaining a lower mean square error between the measured and simulated values; this is known as a nonlinear optimization problem. In our case, the values of the wind effect ventilation coefficient were varied $C_w$, as well as the infiltration coefficient $C_f$, the heat transfer coefficient of the cover $\alpha_c$, and the heat exchange coefficient through the soil $k_s$.

On the other hand, goodness-of-fit parameters such as mean absolute error *(MAE)*, mean square error *(MSE)*, and mean absolute percentage error *(MAPE)* were used to measure the fit of the simulated data with the measured data. Finally, the predictive efficiency *(EF)* of the balance model was determined, and these goodness-of-fit parameters were calculated using the following mathematical expressions:

$$MAE = \frac{1}{m}\sum_{i=1}^{m}|T_{si} - T_{mi}| \tag{17}$$

$$MSE = \frac{1}{m}\sum_{i=1}^{m}(T_{si} - T_{mi})^2 \tag{18}$$

$$MAPE = \left(\frac{MSE}{T_{mi}}\right) * 100 \tag{19}$$

$$EF = 1 - \frac{\frac{1}{m}\sum_{i=1}^{m}(T_{si} - T_{mi})^2}{\frac{1}{m}\sum_{i=1}^{m}(T_{si} - T_{mi})^2} \tag{20}$$

where *i* is the point data, *m* is the number of the total data set, $T_{si}$ and $T_{mi}$ are the simulated and measured temperatures at the point *i*. Once the model was calibrated, a data set obtained between 00:00 h on 23 June and 23:59 h on 26 June was used to implement the mathematical model after the calibration process and to determine its accuracy. Finally, the validity or rejection of the energy balance model was tested by means of a statistical analysis using a hypothesis test for the difference of measurements with homogeneous variances.

## 3. Results and Discussion

*3.1. Behavior of the Input Variables*

### 3.1.1. Temperature

Figure 2 shows the behavior of the temperatures recorded in the soil, cover, and outside environment of the greenhouse during the six days of the experimental period. In general, it was observed that both the soil and the cover react directly to the behavior of the outside environment temperature. Regarding the measured parameters, it was found that the cover temperature presented the highest thermal amplitude, with a mean value of $13.6 \pm 4.4$ °C and maximum and minimum values of 29.9 and 4.9 °C, respectively. In the case of the outdoor environment, the mean temperature was $13.5 \pm 3.0$ °C, with maximum and minimum values of 25.7 and 8.4 °C, respectively. Finally, in the soil, the mean value was $16.5 \pm 1.7$ °C, and the maximum and minimum values were 22.7 and 13.4 °C, respectively. This component of the analyzed biosystem presents less variation on the temporal scale, which agrees with what was reported by Wang et al. [37].

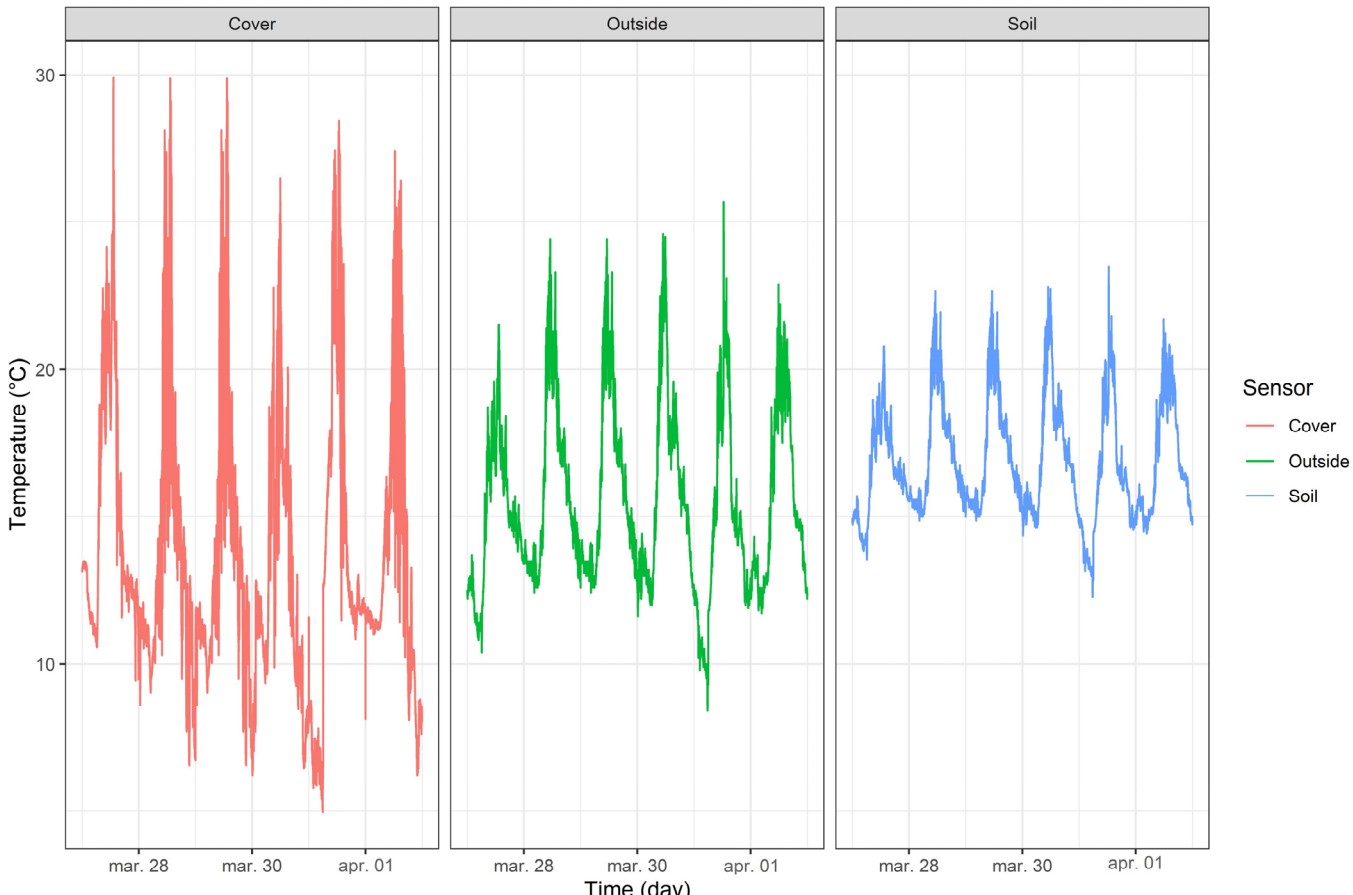

**Figure 2.** Behavior of cover and soil temperatures measured inside the greenhouse and external air temperatures used as inputs to the mathematical model.

3.1.2. Solar Radiation

Figure 3 shows the behavior of solar radiation during the experimental period. In general, it was observed that, during the periods between 18:00 and 6:00 h, solar radiation remained constant at 0 $\mathrm{Wm}^{-2}$. On the other hand, during the periods between 6:00 and 18:00 h, solar radiation showed an upward trend from morning to noon, followed by a downward trend from noon to evening. This type of behavior is characteristic in the savanna of Bogotá, Colombia [38]. Regarding the radiation values recorded, a mean value of 168.4 ± 271.3 $\mathrm{Wm}^{-2}$ was found, with maximum and minimum values of 1120 and 0 $\mathrm{Wm}^{-2}$, respectively. It is important to note that the recording of this variable is relevant since solar radiation is the main energy source that promotes the heating of the greenhouse's internal air; therefore, it is a fundamental component in the different existing greenhouse microclimate simulation models [39].

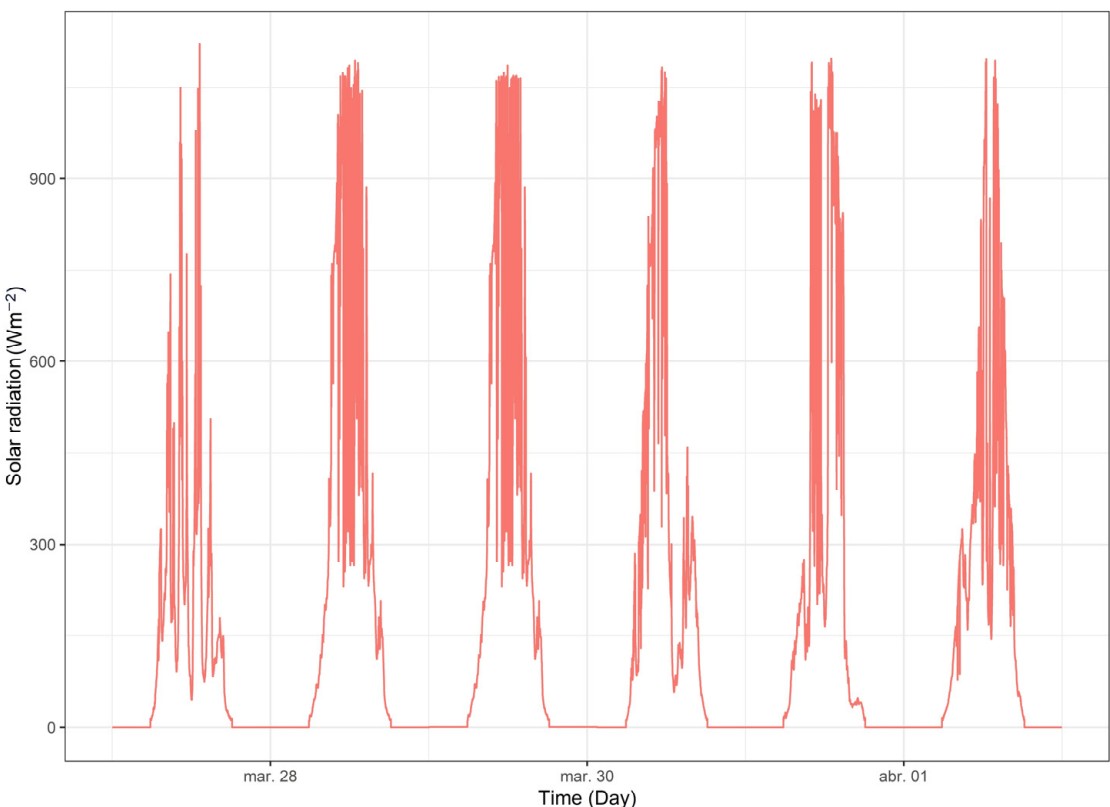

**Figure 3.** Solar radiation behavior outside the greenhouse.

### 3.1.3. Wind Speed

Wind speed in the study region shows an oscillating behavior on the time scale (Figure 4). The mean value of wind speed was $0.7 \pm 0.51$ ms$^{-1}$, while the maximum and minimum values were 3.9 and 0 ms$^{-1}$, respectively. This behavior is characteristic in Colombia and in the Andean region, where the wind presents values above 2 ms$^{-1}$ during the midday hours and values below 1 ms$^{-1}$ during the night hours [40]. It is important to note that wind speed is a major component of natural ventilation, which plays a fundamental role in temperature regulation in passive greenhouses. Therefore, the measurement and recording of this variable are very relevant for the calculation of the analyzed energy balance model [41,42].

### 3.1.4. Relative Humidity

In the outdoor environment, the relative humidity presented a mean value of $77.1 \pm 10.8\%$, and the maximum and minimum values were 97% and 39%, respectively (Figure 5). This behavior is typical of the conditions of a cold and humid climate such as that present in the Bogotá savanna. The recording of this variable is relevant since it is the input variable used to analyze the mass transfer that occurs between the external and internal environment and that influences the behavior of the relative humidity inside the greenhouse and the presence or absence of the condensation of water vapor phenomenon on the internal layer of the greenhouse cover [43,44].

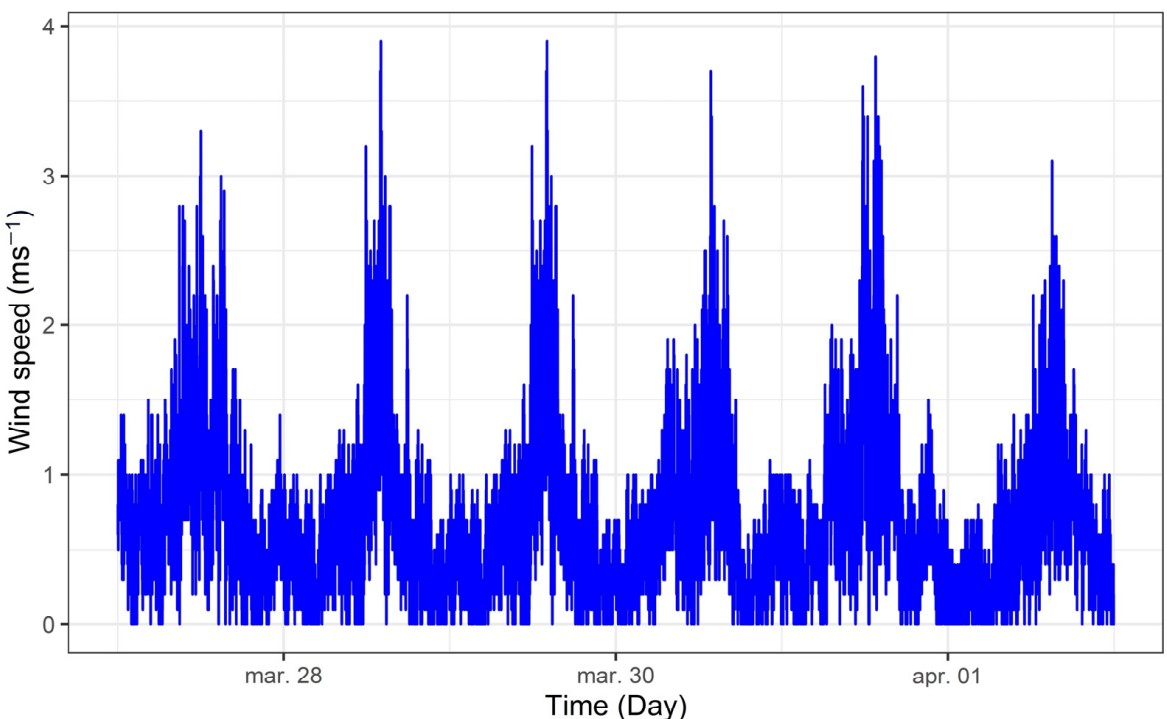

**Figure 4.** Outside wind speed behavior.

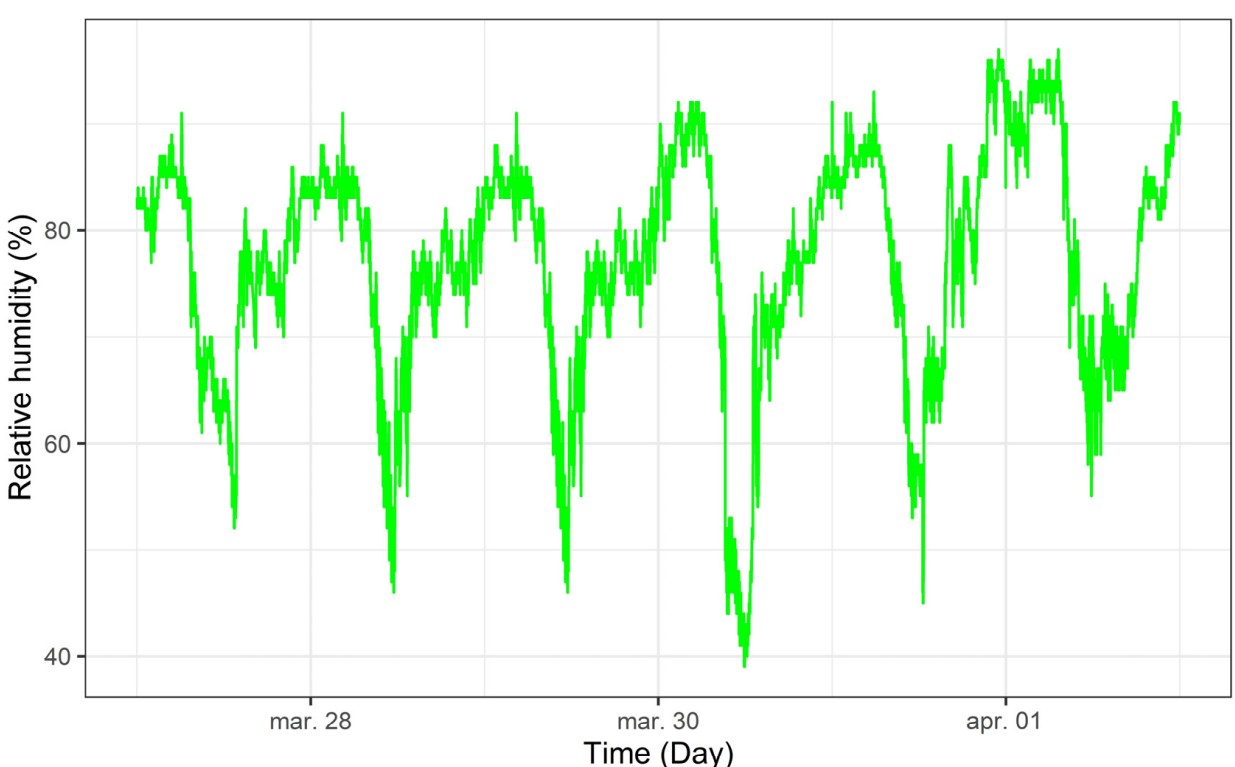

**Figure 5.** Relative humidity behavior of the environment outside the greenhouse.

*3.2. Model Calibration*

Once the model was implemented, a numerical simulation was performed with the input data set and the parameters established in Table 1. Under this configuration, the results obtained for the goodness-of-fit parameters were 1.55 and 3.05 °C for MAE and MSE, respectively, with a MAPE of 17.2% and a prediction efficiency value of 79.8%. These values

can be considered acceptable; however, it is recognized that a calibration process is required to obtain higher accuracy in temperature prediction using the model [45]. Therefore, an optimization process was carried out using the nonlinear least squares technique, varying four parameters of the model that depend on the architecture and roofing material of the greenhouse, between a minimum and maximum range, until finding the final value that offers the best degree of adjustment; these final values are summarized in Table 4.

**Table 4.** Parameters adjusted in the calibration process.

| Initial Value | Reference | Range | Final Value |
|:---:|:---:|:---:|:---:|
| $C_w = 0.15$ | [35] | $0.05 \leq C_w \geq 0.2$ | $C_w = 0.045$ |
| $C_f = 0.0083$ | [36] | $0.0083 \leq C_f \geq 0.023$ | $C_f = 0.011$ |
| $\alpha_c = 5$ | [36] | $1 \leq \alpha_c \geq 10$ | $\alpha_c = 4.3$ |
| $k_s = 5.75$ | [36] | $1 \leq k_s \geq 20$ | $k_s = 7.8$ |

Table 5 shows the results of the calibration process, where the goodness-of-fit parameters were improved. In the case of MAE and MSE, values of 1.13 and 1.83 °C were obtained, respectively. It is important to remember that, the closer these values are to the absolute 0, the greater the predictive capacity of the model; in addition, these values are within the ranges reported in a similar work on a greenhouse with a plastic cover developed by Ghosal and Mishra [46]. Regarding the MAPE, a value of 8.1% was obtained, which is below the recommended maximum limit of 10% for this parameter in greenhouse microclimate prediction models [47,48]. This demonstrates a good fit for the model and its ability to predict accurately. Finally, the prediction efficiency of the model was improved by 13%, obtaining a final value of 0.92, which, being a value close to 1, guarantees that the simulated data are very similar to the real data; therefore, the model can be considered a good predictor of the simulated process (Table 5).

**Table 5.** Goodness-of-fit parameters obtained for the mathematical model in the calibration phase.

| Measure of Adjustment | Before Calibration | After Calibration |
|:---|:---:|:---:|
| MAE | 1.55 | 1.13 |
| MSE | 3.05 | 1.83 |
| MAPE | 17.2 | 8.1 |
| EF | 0.79 | 0.92 |

Table 6 shows the results obtained from the statistical analysis performed to determine the validity or rejection of the energy balance model as an adequate predictor of the greenhouse temperature analyzed. Once the normality of the measured and simulated data sets was verified, a hypothesis test was performed to determine whether the variances of the data sets were homogeneous. The value of the F contrast statistic was calculated for a significance level of 0.05, obtaining an F value equal to 0.961, while the *p*-value was 0.771 for the temperature values. These results suggest that the p-value was much higher than the significance level; therefore, the null hypothesis cannot be rejected. Therefore, the variances of the data can be considered to be equal ($\sigma_{(Dm)}^2 = \sigma_{(Ds)}^2$). Finally, a confidence interval with values of {0.692, 1.041} was obtained. This confidence interval provides a range of possible values for the true variance ratio. In this case, the interval includes 1, which supports the idea that the variances are statistically similar. Therefore, considering that these results from the measured and simulated data are statistically similar, the simulation model can be accepted.

**Table 6.** Test statistic results for the model.

| F Test to Compare Two Variances | $H_0$: $\sigma_{(Dm)}^2 = \sigma_{(Ds)}^2$ o $H_1$: $\sigma_{(Dm)}^2 \neq \sigma_{(Ds)}^2$ |
|---|---|
|  | Temperature |
| F | 0.961 |
| *p*-value | 0.771 |
| 95% confidence interval | [0.692, 1.041] |
| The null hypothesis ($H_0$) is accepted. | |

### 3.3. Simulation of Temperature Behavior Inside the Greenhouse with Calibration Data

Figure 6 shows the temporal variation of the temperature inside the greenhouse during the six days of experimental measurements. In general, it can be observed that the energy balance model proposed has a high capacity to predict the behavior and trend of the temperature inside the greenhouse: the model predicts in an acceptable way the high-temperature peaks that occur during the hours of high radiation and the cooling that occurs during the night hours. It is also important to mention that the energy balance model proposed responds adequately to changes in the external climatic variables, which are considered fast variables that, in tropical climates, change their value in small time scales of even seconds or minutes, which, in turn, generates changes in the microclimate of the greenhouse [49].

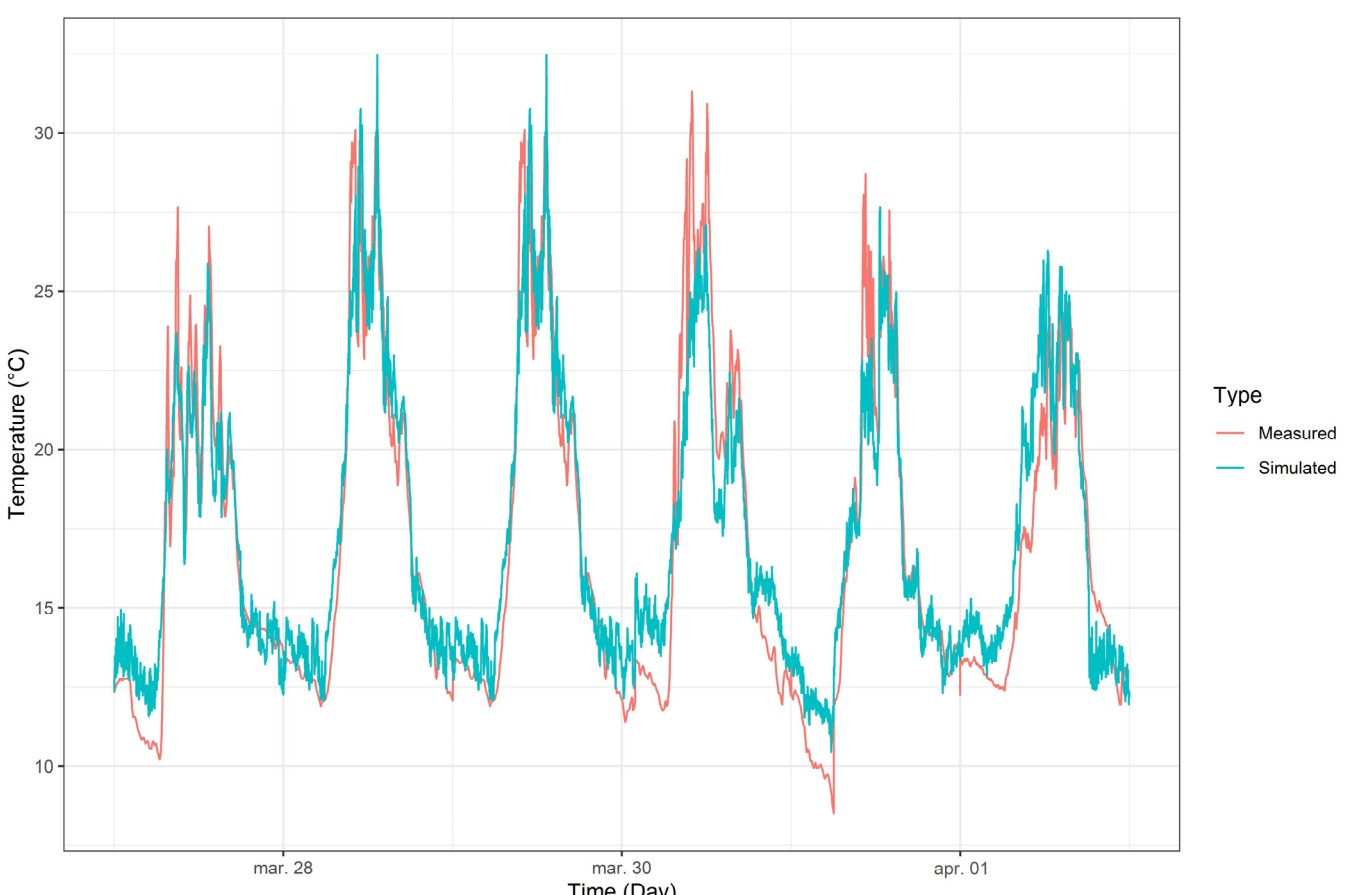

**Figure 6.** Temporal behavior for the simulated and measured temperature inside the greenhouse in the calibration phase.

However, it is necessary to mention that the model tends to overestimate the value of the temperature during the night hours, which can be improved with future studies in which it is possible to model and quantify in a more precise way the loss of far infrared

radiation that occurs during the night in this type of greenhouse with a plastic cover, as this is the cause of the drastic reduction in temperature inside the greenhouse [50]. This can also be complemented with models that include a multi-layer analysis of the thermal behavior of the soil and the accurate transpiration of the crops to be grown [10].

As for the behavior of the temperature inside the greenhouse during daytime conditions (6:00 to 18:00 h), the mean temperature for the six days was $20.7 \pm 3.4$ °C, and this value ranged between $19.1 \pm 3.5$ °C and $22.2 \pm 4.1$ °C (Table 7). On the other hand, the maximum temperature presented a mean value of 28.7 °C with values ranging between 25.9 and 32.4 °C; moreover, the mean minimum temperature presented a value of 13.2 °C, and its values ranged between 11.1 and 14 °C, respectively. During the night period (18:00 to 6:00 h), a mean temperature of $13.7 \pm 0.9$ °C was recorded, with values varying between 12.9 and 14.3 °C; likewise, the mean maximum temperature was 16.8 °C, with values varying between 14.9 and 18.5 °C, respectively. On the other hand, the minimum temperature presented a mean value of 11.3 °C, with values ranging between 10.3 °C and 12.3 °C. These values predicted by the mathematical model, both for the night and daytime periods, are within the ranges reported in various experimental works carried out in plastic-covered greenhouses established in the Bogotá savanna [51–55].

**Table 7.** Simulated temperature values obtained inside the greenhouse during the day and night periods in the calibration phase.

| Date Day/Month/Year | Period Hour | Simulated Temperature (°C) | | |
|---|---|---|---|---|
| | | $T_{mean}$ | $T_{maximum}$ | $T_{minimum}$ |
| 27 March 2023 | From 00:00 to 06:00 | $12.9 \pm 0.6$ | 14.9 | 11.6 |
| 27 March 2023 | From 06:00 to 18:00 | $19.6 \pm 2.6$ | 25.9 | 12.4 |
| 27 March 2023 | From 18:00 to 06:00 | $13.7 \pm 0.7$ | 15.9 | 12.1 |
| 28 March 2023 | From 06:00 to 18:00 | $22.1 \pm 4.1$ | 32.2 | 13.9 |
| 28 March 2023 | From 18:00 to 06:00 | $13.9 \pm 1.0$ | 18.5 | 12.1 |
| 29 March 2023 | From 06:00 to 18:00 | $22.2 \pm 4.2$ | 32.4 | 13.9 |
| 29 March 2023 | From 18:00 to 06:00 | $14.3 \pm 0.9$ | 18.4 | 12.1 |
| 30 March 2023 | From 06:00 to 18:00 | $20.3 \pm 3.1$ | 27.1 | 13.8 |
| 30 March 2023 | From 18:00 to 06:00 | $13.6 \pm 1.7$ | 16.5 | 10.4 |
| 31 March 2023 | From 06:00 to 18:00 | $19.1 \pm 3.5$ | 27.6 | 11.1 |
| 31 March 2023 | From 18:00 to 06:00 | $13.9 \pm 0.6$ | 16.7 | 12.3 |
| 1 April 2023 | From 06:00 to 18:00 | $20.6 \pm 3.1$ | 26.2 | 14.0 |
| 1 April 2023 | From 18:00 to 00:00 | $13.3 \pm 0.8$ | 16.7 | 10.3 |

Regarding the temperature of the outdoor environment, an interesting behavior can be observed in Figure 7. In general, it is observed that, during the daytime hours (6:00 to 18:00 h), the value of this parameter shows the thermal gain caused by the incident solar radiation in the greenhouse, with a mean value of $3.2 \pm 1.4$ °C and a maximum and minimum value of 9.8 and 1.2 °C, respectively. Meanwhile, in the evening hours (18:00 to 6:00 h), the mean value of the differential was $0.3 \pm 0.6$ °C, with maximum and minimum values of 2.9 and $-2.8$ °C, respectively.

It is important to note that, on some nights, the phenomenon of thermal inversion was observed, wherein the air inside the greenhouse is colder than the outside air. This is a characteristic phenomenon of Colombian greenhouses with a plastic cover and occurs due to the low capacity of the plastic to retain the thermal radiation generated from the greenhouse floor [56,57]. Therefore, these results continue to demonstrate the high predictive capability of the mathematical model to simulate the thermal behavior of a Colombian greenhouse.

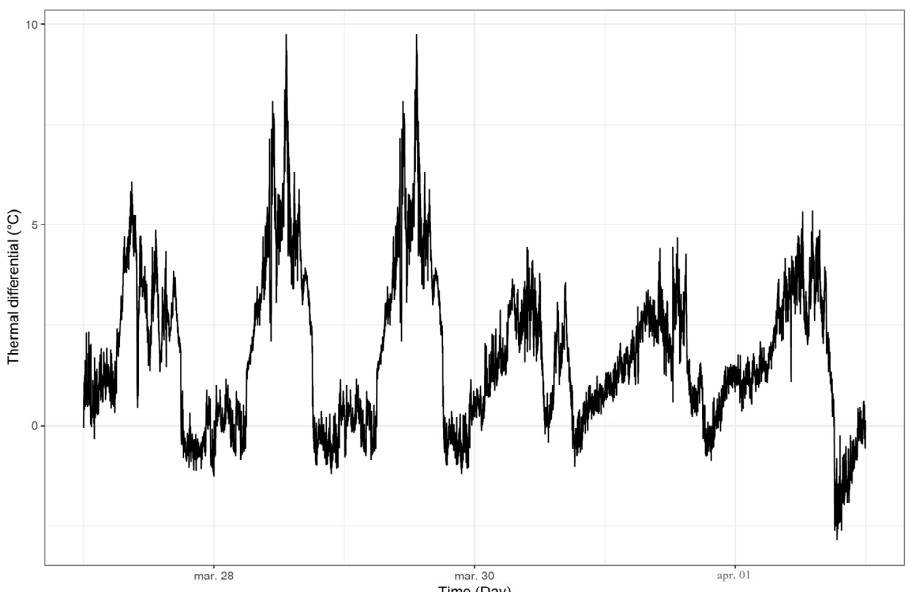

**Figure 7.** Thermal differential between the inside and outside environment of the greenhouse obtained using a simulation in the calibration phase.

### 3.4. Temperature Behavior in the Model Implementation Phase

To show the robustness of the model, a different data set than the one used in the calibration phase was selected. The simulated and measured temperature for the period between 21 and 25 June is shown in Figure 8. In general, it can be observed that the model has a high prediction capacity of the behavior of the temperature value trends both in the day and night climate periods. It predicts the high temperatures that are characteristic of Colombian greenhouses at midday and the low temperatures that occur at dawn, as has already been demonstrated in multiple works [58]. The mean value of the absolute error of the prediction was $1.08 \pm 1.16$ °C for the whole data set, which can be considered an adequate value that is within the ranges obtained in works like this research.

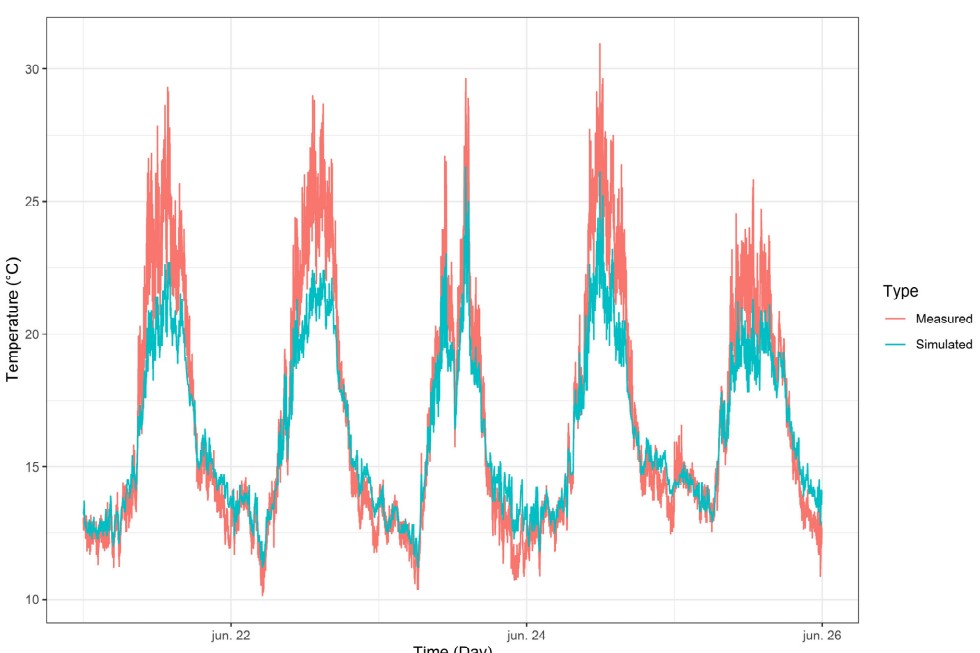

**Figure 8.** Temporal behavior for simulated and measured temperature inside the greenhouse in the implementation phase.

The goodness-of-fit parameters for the data collected for the implementation phase resulted in values of 1.39 and 2.13 °C for MAE and MSE and 9.4% for MAPE. These values, while not as precise as in the calibration phase, are still acceptable and demonstrate the model's ability to predict the greenhouse's thermal behavior effectively. Furthermore, the model's prediction efficiency reached 85% (Table 8), reinforcing its accuracy. This research affirms that the implemented model is simple yet reliable, offering valuable insights into the processes governing the thermal behavior of plastic cover greenhouses [59]. Moreover, its ease of implementation facilitates the optimization and control of greenhouse microclimates [49].

**Table 8.** Goodness-of-fit parameters obtained for the mathematical model in the implementation phase.

| Measure of Adjustment | Implementation Phase |
|---|---|
| MAE | 1.39 |
| MSE | 2.13 |
| MAPE | 9.4 |
| EF | 0.86 |

## 4. Conclusions

In this study, a simple dynamic energy balance model was calibrated and implemented to understand the physical processes involved in the heating and cooling of a plastic-covered greenhouse built in a tropical high-mountain climate.

The developed model considered the parameters dependent on the architecture and geometry of the greenhouse, which allows its adaptation to any type of Colombian greenhouse. It was observed that soil and cover temperature, as well as ventilation rate, significantly influenced the prediction of the greenhouse's internal temperature. A calibration process involved adjustments to parameters related to these aspects, leading to an enhancement in the model's prediction efficiency to 92%, with a MAPE of 8.1%. Furthermore, these parameters reached values of 86% and 9.4% in the implementation phase, indicating the model's strong predictive capability.

The implemented model was able to predict the thermal inversion phenomena that are characteristic of Colombian greenhouses and predicted temperatures ranging between 10.3 and 32.4 °C. These results are within the ranges reported in previous experimental studies developed in different types of greenhouses under similar climatic conditions.

The results obtained in this work can form the basis for future research that seeks to quantify and optimize the energy consumption of this greenhouse for purposes applicable to air conditioning activities such as heating and cooling, either through active or passive methods. Finally, this work serves as a basis for studies that seek to strengthen the mathematical model applied to other greenhouse types with the objective of improving its predictive capacity and, thus, being able to have a very reliable tool for decision-making.

**Author Contributions:** Conceptualization, E.V., G.A.O. and A.N.C.; methodology, E.V., G.A.O., A.N.C. and I.L.L.-C.; software, G.A.O. and A.N.C.; validation, E.V., G.A.O., A.N.C. and I.L.L.-C.; formal analysis, E.V., G.A.O., A.N.C., I.L.L.-C. and J.F.A.-C.; investigation, E.V., G.A.O., A.N.C., I.L.L.-C. and J.F.A.-C.; resources, E.V., G.A.O., A.N.C., I.L.L.-C. and J.F.A.-C.; data curation, E.V. and I.L.L.-C.; writing—original draft preparation, E.V., G.A.O., A.N.C., I.L.L.-C. and J.F.A.-C.; writing—review and editing, E.V., G.A.O., A.N.C., I.L.L.-C. and J.F.A.-C.; visualization, E.V., G.A.O., A.N.C. and I.L.L.-C.; supervision, E.V.; project administration, E.V.; funding acquisition, E.V. All authors have read and agreed to the published version of the manuscript.

**Funding:** This study was financed by the Ministry of Science, Technology and Innovation of Colombia—MINCIENCIAS—through a project called "Fortalecimiento de las capacidades de I + D + i del centro de investigación Tibaitata para la generación, apropiación y divulgación de nuevo conocimiento como estrategia de adaptación al cambio climático en sistemas de producción agrícola ubicados en las zonas agroclimáticas del trópico alto colombiano".

**Data Availability Statement:** The data are contained in the article.

**Acknowledgments:** The authors would like to thank the Corporación Colombiana de Investigación Agropecuaria—AGROSAVIA for their technical support in carrying out this research.

**Conflicts of Interest:** The authors declare no conflict of interest.

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
