# Peer review of "Calibration and Implementation of a Dynamic Energy Balance Model to Estimate the Temperature in a Plastic-Covered Colombian Greenhouse"

_agriengineering, doi:10.3390/agriengineering5040140_

Round 1

Reviewer 1 Report

Comments and Suggestions for Authors

1)Please give how to distribute the sensors? also give the sensor accuracy.

2) It is a interesting paper, I suggest to accept this paper after minor revision.

Author Response

Dear Reviewer.

Reviwer 1.

Please give how to distribute the sensors? also give the sensor accuracy.

Response: The following text was included in the new version of the manuscript:

Inside the greenhouse, an Aranet sensor and a data logging station were used to record air temperature. In addition, two sensors were used to measure and record soil and greenhouse cover temperature data, which were connected to an application generated using the Arduino open-source electronic prototyping platform. The technical specifications of these sensors are detailed in Table 3. The air and soil temperature sensors were located in the central zone of the greenhouse at 1.0 m above ground level and 0.1 m deep respectively, while the greenhouse cover sensor was located in the central zone just above the middle cross section of the roof. Additionally in table 3, all sensor information is included.

It is a interesting paper, I suggest to accept this paper after minor revision.

Response: The revisions were incorporated, appreciated reviewer thank you very much for contributing to the improvement of the document.

Regards

The authosr

Reviewer 2 Report

Comments and Suggestions for Authors

Comprehensive paper should go for publication. 

Comments on the Quality of English Language

Requires minor edits

Author Response

Dear Reviewer

Thank you very much appreciated reviewer for your review and concept.

Regards

The authosr

Reviewer 3 Report

Comments and Suggestions for Authors

Comments 

1. Abstract should be more constructive and informative.

2. In Introduction, longer sentences sentences should be split into two statements.

3. In this Ms., authors have cited nearly 15 references of same author group EA Villagran. Only most relevant references should be cited from other researchers also.

4. Novelty of this research work should be included.

5. Authors should provide more clear Figures.

6. All experimental findings should be statistically validated for better presentations wherever applicable.

7. References must be cross checked.

8. Latest references of other researchers should be cited.

Comments on the Quality of English Language

Need an English expert to check this

Author Response

Dear reviewer, thank you very much for your suggestions, which were included in the new version of the manuscript.

Abstract should be more constructive and informative.

Response: The abstract was modified as follows:

Modeling and simulation have become fundamental tools for the microclimatic analysis of greenhouses under various climatic conditions. These models allow a precise control of the climate inside the structures and the optimization of their performance under any situation. In Colombia, the availability of energy balance models adapted to local greenhouses and their climate is limited, which affects the decision making of both technical advisors and growers. This study focused on calibrating and evaluating a dynamic energy balance model to predict the thermal behavior of an innovative type of plastic cover greenhouse designed for the Bogotá savanna. The selected model considers fundamental heat and mass transfer processes, incorporating parameters that depend on the architecture of the structure and local climatic conditions, which makes it suitable for protected agriculture in Colombia. The results of the post-calibration evaluation showed that the model is highly accurate, with a temperature prediction efficiency close to 86%. This ensures that the model can accurately predict the thermal behavior of the greenhouse being evaluated. It is important to note that the model can also anticipate phenomena characteristic of Colombian greenhouses, such as thermal inversion. This advance becomes a valuable tool for decision making in protected agriculture in the region.

In Introduction, longer sentences sentences should be split into two statements.

The changes were applied in several paragraphs of the introduction and are highlighted in green.

In this Ms., authors have cited nearly 15 references of same author group EA Villagran. Only most relevant references should be cited from other researchers also.

Five references have been eliminated and only those most relevant to the context of the region where the study was conducted have been left.

Novelty of this research work should be included.

Response: The following text was included in the new version of the manuscript:

The novelty of this work is that it includes equations that allow modeling the management of the lateral and roof ventilation areas traditionally used in Colombia. In addition, the model, being versatile, can be implemented in the different types of Colombian inveranderos, just by making some changes in parameters related to the geometry of the structure to be analyzed.

Authors should provide more clear Figures.

Response: Figures were improved in resolution, to make them more legible to the reader.

All experimental findings should be statistically validated for better presentations wherever applicable.

Response: The degree of fit and accuracy of the model was evaluated through goodness-of-fit parameters, this type of comparisons between experimental and simulated data are the ones that are usually applied in this type of investment. However, we thank the reviewer for his suggestion and will take it into account for future work related to this research.

References must be cross checked.

Response: References have been collated and left in the journal format.

Latest references of other researchers should be cited.

Response: The following references were included:

Rasheed, A., Kim, H. T., & Lee, H. W. (2022). Modeling-based energy performance assessment and validation of air-to-water heat pump system integrated with multi-span greenhouse on cooling mode. Agronomy12(6), 1374.

Dimitropoulou, A. M. N., Maroulis, V. Z., & Giannini, E. N. (2023). A Simple and Effective Model for Predicting the Thermal Energy Requirements of Greenhouses in Europe. Energies16(19), 6788.

Calise, F., Cappiello, F. L., Cimmino, L., & Vicidomini, M. (2023). Dynamic Modelling and Energy, Economic, and Environmental Analysis of a Greenhouse Supplied by Renewable Sources. Applied Sciences13(11), 6584.

Lee, C. G., Cho, L. H., Kim, S. J., Park, S. Y., & Kim, D. H. (2022). Prediction Model for the Internal Temperature of a Greenhouse with a Water-to-Water Heat Pump Using a Pellet Boiler as a Heat Source Using Building Energy Simulation. Energies15(15), 5677.

Faniyi, B., & Luo, Z. (2023). A Physics-Based Modelling and Control of Greenhouse System Air Temperature Aided by IoT Technology. Energies16(6), 2708.

Reviewer 4 Report

Comments and Suggestions for Authors

This is a unique paper on climate models. Please add explanations for the following.

â‘  This research is about microclimate analysis, but what range (area) is the "microclimate" supposed to be? â‘¡The results of this research show that the temperature prediction efficiency is approximately 86%, but what is the reason why the remaining 14% could not be predicted? Please give the possible reason. â‘¢ How was the plan for the greenhouse model (3-4 pages) decided? Please briefly explain the background and process.

Author Response

Dear reviewer, thank you very much for your review, it has helped to improve the document.

This is a unique paper on climate models. Please add explanations for the following.

This research is about microclimate analysis, but what range (area) is the "microclimate" supposed to be?

Response: In work on greenhouse microclimate, the work is only applicable to the area and volume of air covered by the greenhouse.

The results of this research show that the temperature prediction efficiency is approximately 86%, but what is the reason why the remaining 14% could not be predicted? Please give the possible reason.

Response: The following text is included in the manuscript:

However, it is necessary to mention that the model tends to overestimate the value of the temperature during the night hours, which can be improved with future studies where it is possible to model and quantify in a more precise way the loss of far infrared radiation that occurs during the night in this type of greenhouses with plastic cover and that is the cause of the drastic reduction of the temperature inside the greenhouse [50]. This can also be complemented with models that include a multi-layer analysis of the thermal behavior of the soil and also accurately include the transpiration of the crops to be grown [10].

How was the plan for the greenhouse model (3-4 pages) decided? Please briefly explain the background and process.

Response: In the introduction and methodology section, the background and previous works developed with this model are mentioned, and some other references where works with this general energy balance model used in greenhouses are carried out are also mentioned and included in the introduction.

Grettings

The authors

Round 2

Reviewer 3 Report

Comments and Suggestions for Authors

Author replied all the comments suitably. So, I recommend for possible consideration of publication of this Ms.

Author Response

Thanks